# Robot-Assisted versus Laparoscopic Gastrointestinal Surgery: A Systematic Review and Metanalysis of Intra- and Post-Operative Complications

**DOI:** 10.3390/jpm13091297

**Published:** 2023-08-25

**Authors:** Carmine Iacovazzo, Pasquale Buonanno, Maria Massaro, Marilena Ianniello, Andrea Uriel de Siena, Maria Vargas, Annachiara Marra

**Affiliations:** Department of Neurosciences, Reproductive and Odontostomatological Sciences, University of Naples, Federico II, 80131 Naples, Italy; pasquale.buonanno@unina.it (P.B.); marilena.ianniello@gmail.com (M.I.); andreauriel@outlook.it (A.U.d.S.);

**Keywords:** minimally invasive surgery, surgery complications, robotic-assisted surgery, gastric, colorectal

## Abstract

Background: The use of robotic surgery is attracting ever-growing interest for its potential advantages such as small incisions, fine movements, and magnification of the operating field. Only a few randomized controlled trials (RCTs) have explored the differences in perioperative outcomes between the two approaches. Methods: We screened the main online databases from inception to May 2023. We included studies in English enrolling adult patients undergoing elective gastrointestinal surgery. We used the following exclusion criteria: surgery with the involvement of thoracic esophagus, and patients affected by severe heart, pulmonary and end-stage renal disease. We compared intra- and post-operative complications, length of hospitalization, and costs between laparoscopic and robotic approaches. Results: A total of 18 RCTs were included. We found no differences in the rate of anastomotic leakage, cardiovascular complications, estimated blood loss, readmission, deep vein thrombosis, length of hospitalization, mortality, and post-operative pain between robotic and laparoscopic surgery; post-operative pneumonia was less frequent in the robotic approach. The conversion to open surgery was less frequent in the robotic approach, which was characterized by shorter time to first flatus but higher operative time and costs. Conclusions: The robotic gastrointestinal surgery has some advantages compared to the laparoscopic technique such as lower conversion rate, faster recovery of bowel movement, but it has higher economic costs.

## 1. Introduction

Robotic surgery is attracting ever-growing interest as it allows surgeons to have a three-dimensional vision of the operating field, fine movements of robotic arms, and smaller incisions, leading to many advantages such as reduced blood loss, lower risk of damage of nervous, vascular, and parenchymal structures, and less post-operative pain, but it is also associated with an extreme position of the patient that can affect the incidence of some side effects, such as atelectasis [1,2,3]. Even if robotic surgery shares many characteristics with the standard laparoscopic technique, it overcomes some drawbacks of laparoscopy such as the restricted range of motion of instruments, poorly ergonomic position of the surgeon, and the two-dimensional view [1,4].

Gastrointestinal (GI) surgery is one of the most important fields of application of robotic systems and the number of procedures performed with this approach is rapidly increasing despite the doubtful relationship between costs and benefits [5]. During the last 50 years, the use of minimally invasive surgery spread all over the world, and video-laparoscopic-assisted surgery became a common technique; a recent study reported that the average number of the laparoscopic abdominal GI procedures performed by residents increased considerably in the last decades [6]. Even if laparoscopic-assisted surgery largely demonstrated its advantages, the incidence of complications (i.e., pneumonia, ICU recovery, stroke, acute kidney failure, cardiac arrest, anastomotic leakage, deep vein thrombosis, and other respiratory or cardiovascular complications) in major GI surgery remained high (33–44%), thus worsening the prognosis and the quality of life of the patients and increasing the length of hospitalization and the healthcare-related costs [7,8,9]. The wide diffusion of laparoscopic systems was probably due to the relative low costs of the system (about 1000£/patient for disposables and about 90£/patient for system) that seems to be well balanced by the several advantages of this technique [10].

The development of robot systems started in the 1990s, and in last two decades, various robot models were produced, such as HUGO^®^ (Medtronic^©^, Minneapolis, MN, USA), Micro Hand S^®^ (Wego Surgical Robot Co. LTD., Weihan, China), Telelap ALF-X^®^ (SOFAR—Surgical Robotics Division^©^, Trezzano Rosa, Italy), and Da Vinci^®^ (Intuitive Surgical Co. ^©^, Sunnyvale, CA, USA), which is the most widespread device used in gastrointestinal surgery [11]. Nowadays, even if the robotic approach has several advantages, the prevalence of the use of the gastrointestinal robot-assisted technique is only 7.8% against the 36% and 46.8% of open and laparoscopic approaches, respectively, which is probably because of the cost which limits its diffusion in all the hospitals and the need for a specific training for the surgeons [12]; indeed, the costs of the purchasing a robot system is about GBP 1.7 million, the maintenance costs are about GBP 140,000 per year, and the disposables cost GBP 1000 per patient [13]. Robotic-assisted surgery seems to be related to a lower incidence of conversion rate to open surgery compared to video laparoscopy, but it is unclear if it could reduce the length of hospitalization, the incidence of intra- and post-operative complications, and the estimated blood loss [3,12,14]. Therefore, the robotic surgery often adopts a forced Trendelenburg or anti-Trendelenburg position, which could increase the incidence of many intra- and post-operative complications [15,16]. The difference between the minimally invasive approaches and open surgery was investigated by many randomized controlled trials (RCTs), but few RCTs were conducted to compare laparoscopic and robotic-assisted techniques in gastrointestinal surgery.

The aim of this work was to investigate the impact of robot systems on post-operative complications, length of hospitalization, estimated blood loss, quality of life, conversion to open surgery rate, and economic costs in comparison to the video-laparoscopic-assisted technique in elective gastrointestinal surgery. To our knowledge, this is the first systematic review and metanalysis that investigated the incidence of intra- and post-operative complications between the laparoscopic and robotic-assisted approach. The different incidence of pneumonia of robotic-assisted surgery should be considered in the decision-making process to evaluate which approach could be used, especially in patients with high risk factors for pneumonia undergoing elective GI surgery. Moreover, the clinical impact of the lower rate of conversion to open surgery and the faster recovery of the bowel motility of the robotic surgery are unclear, and more studies are needed to define the real advantages and costs of the robotic-assisted GI surgery.

## 2. Materials and Methods

This systematic review and meta-analysis was conducted according to Preferred Reporting Items for Systematic Reviews and Meta-Analyses (PRISMA) [17]. It was registered on the PROSPERO database with the following registration number: CRD42023437060.

Extensive research was undertaken in the following databases from their inception until 30 May 2023: Medline, Cumulative Index to Nursing and Allied Health Literature (CINAHL), Web of Science, and Cochrane Central Register of Controlled Trials (CENTRAL). It was used the following string of research: << (((((((((((((((((((((((((robot-assisted) OR (laparoscopic-assisted)) OR (video-laparoscopic-assisted)) OR (robotic)) AND (surgery)) OR (surgical procedure)) AND (stomach)) OR (gastrectomy)) OR (fundoplication)) OR (small intestine)) OR (large intestine)) OR (colectomy)) OR (anastomosis)) OR (rectal)) OR (colic)) OR (colorectal)) OR (hernia)) OR (liver)) OR (hepatectomy)) OR (pancreas)) OR (pancreatectomy)) OR (Whipple procedure)) OR (Whipple’s procedure)) OR (gallbladder)) OR (cholecystectomy)) OR (small bowel resection) >>.

We used the following inclusion criteria: randomized controlled trials (RCTs) comparing video-laparoscopic-assisted surgery and robot-assisted surgery in patients aged over than 18 years old undergoing elective gastrointestinal surgery. We used the following exclusion criteria: patients undergoing thoracic esophagus surgery; patients affected by mental or neurological disorders that could compromise the expression of the written consent; patients affected by chronic heart failure with a NYHA score III–IV; patients with one lung; patients with any previous transplanted organ; patients affected by end-stage renal disease needing dialysis; patients that were candidates for multiple surgical procedures in the same intervention; observational studies, letters; only abstract studies; articles not written in English. 

Two authors (CI and MM) independently evaluated the matching of inclusion and exclusion criteria based on titles, abstracts, titles, abstracts, and full-text reports; disagreements were solved by discussion.

Data extraction was performed, and an electronic sheet was created to summarize the data. Two researchers (PB and MI) independently assessed the risk of bias of each study using the Cochrane Risk of Bias (RoB2) tool [18]. The researchers evaluated the following domains: randomization process, deviation from the intended interventions, missing outcome data, measurement of the outcome, and selection of the reported result. An overall judgment of risk of bias was assigned among the next three levels: ‘high’, ‘some concerns’, and ‘low’. If it needed, disagreement was solved by consensus.

The primary outcomes were post-operative complications. Post-operative complications were defined in the subsequent way: anastomotic leakage, that is a defect of the anastomotic site that creates a communication between the inner and external compartment; cardiovascular complications such as acute myocardial ischemia, acute heart failure, or hypotension that required intensive care; hospital readmission, defined as the readmission to the hospital due to any complications linked to the surgery; deep vein thrombosis; and respiratory complications defined as pneumonia or lung failure that required admission to intensive care unit.

The secondary outcomes were: conversion to open surgery rate; operative time (defined as the minutes by the start of the preparation of the surgical field after anesthesia until the end of the surgical procedure); length of hospitalization (defined as the days after the surgical procedure to the discharge of patient); time to first flatus (defined as the time after the surgery until to the first flatus); estimated blood loss (defined as the result of multiplying the perioperative difference in hemoglobin or hematocrit with estimated blood volume of the patient) [19]; economic costs (in dollar); mortality rate in next 30 days after surgery; quality of life after 1 month measured with any appropriate scale; and pain score at one day and one month after surgery, which was measured with the numeric rating scale (NRS) or visual analogue scale (VAS).

Continuous variables were presented as mean and standard deviation (SD); mean differences (MD) were used to compare continuous variable while standardized mean differences (SMD) were used to compare continuous data of different scales for quality of life [20]. Dichotomous variables were reported as proportions; the odds ratio (OR) was calculated to evaluate dichotomous variables. All differences were assessed as statistically significant only if the *p*-value < 0.05. An I^2^ statistic was performed to evaluate the heterogeneity of the studies in each analysis. Heterogeneity evaluates the clinical and methodological variety between the analyzed studies, and if the *p*-value is less than 0.05, it indicates a significative heterogeneity; the presence of a significative heterogeneity reduces the generalizability of the results. According to the value of the I^2^ statistic, it is possible to classify the heterogeneity in the following way: ‘low’ if I^2^ is less than 25%; ‘moderate’ if I^2^ ranges between 25% and 50%; ‘high’ if I^2^ is over than 75% [21]. After the first overall analysis, a sensitivity analysis was conducted by excluding studies valued with a ‘high’ risk of bias. Finally, all studies were divided by the type of surgery, and a subgroup analysis was performed for each outcome and each surgery. 

Fragility indexes (FI) were calculated to measure the robustness of each dichotomous outcome in each study. A descriptive statistic with the median, minimum, and maximum were performed to evaluate the robustness of the overall analysis and each study for each outcome. A study had intrinsic robustness when FI is over than 8 [22].

The overall analysis, subgroup and sensitivity analyses, and FI were calculated using R studio package [23]. 

A trial sequential analysis (TSA) of overall and subanalysis was executed for each outcome to establish if the metanalysis was conclusive or more studies are needed. The TSA is a calculation performed to weigh type I and type II errors and graphically establish if the result is strong enough to be unchanged by further studies. The report of a TSA analysis is a graph plotting the following data: cumulative sample size on the *x*-axis that indicates the required information size, which is defined as the number of patients and events needed to detect a priori the intervention effect; cumulative z-score on the *y*-axis; conventional boundaries, which are two horizontal lines that delimit the conventional values for the statistical significance of z-score (i.e., −1.96 and +1.96, corresponding to a *p*-value = 0.05); monitoring boundaries, two ascending or descending lines at the upper and lowest edges of the graph—if the cumulative z-score crosses these lines, it indicates the harm or the benefit of the intervention; futility boundaries, which are two diagonal lines that start from the *x*-axis—if the cumulative z-scores stop in this zone, the results are not significant and are not influence by further studies [24]. The TSA is performed using ‘TSA software-v. 0.9.5.10′ [24]. 

## 3. Results

Figure 1 reports the PRISMA flowchart. After the identification and the screening of the database, 22 studies were assessed for eligibility. Two studies were excluded as they were written in languages other than English (one written in Spanish and one written in Chinese), and one study was excluded because it did not report data relating to the outcomes considered in this study. Finally, 18 studies met the inclusion criteria with a total sample size of 3262 (1642 laparoscopic surgery and 1620 robot-assisted surgery). Appendix A shows the characteristics of the included studies. The studies were divided according to the specific surgery they treated: one colon procedure [25]; three hernia repair surgeries [26,27,28]; nine rectal surgeries [29,30,31,32,33,34,35,36,37]; two Nissen’s fundoplication procedures [38,39]; two gastrectomy procedures [9,40]; and one cholecystectomy [41]. 

Figure 2 illustrates the risk of bias evaluation. Two studies were evaluated as ‘high’ risk of bias due to the high risk of the randomization process; six studies were assessed with ‘some concerns’ due to the randomization process; eleven studies were rated as having a ‘low’ risk of bias. 

### 3.1. Analysis of Intra- and Post-Operative Complications

Figure 3 reports the analysis of all the post-operative complications rates. The risk of pneumonia is higher in the laparoscopic group (N = 519, OR = 0.37, 95% C.I. 0.14/0.98, *p* = 0.042, I^2^ = 12.535%, Het. *p*-value = 0.285). This analysis was performed in only the gastrectomy subgroup due to the absence of data for other types of surgery. No statistically significative differences were found in anastomotic leakage, readmission rate, deep vein thrombosis, cardiovascular complications, and other respiratory complications.

Appendix A shows the sensitivity analysis of post-operative complications. The sensitivity analysis of post-operative complications reports that the anastomotic leakage and readmission rate have no differences between the groups. Appendix A reports the subgroup analysis of intra- and post-operative complications for every type of surgery; the overall complications rate (N = 2076, OR = 0.49, 95% C.I. 0.28/0.87, *p* = 0.014, I^2^ = 0.00%, Het. *p*-value = 0.927) results are lower when robotic surgery is performed in the gastrectomy subgroup: this result is essentially due to the positive effect on post-operative pneumonia recorded in the robotic surgery group, as the rate of other post-operative complications was not significantly different between the robotic and laparoscopic approaches. There are no differences in the specific and overall complications rate in hernia repair and rectal surgeries.

### 3.2. Analysis of Secondary Outcomes

Figure 4 reports the analysis of conversion to open surgery rate, operative time, length of hospitalization, and time to first flatus. The analysis of the conversion to open surgery (Figure 4a) showed that the laparoscopic-assisted approach has a higher risk of conversion to open surgery than robotic technique (N = 2789, OR = 0.56, 95% C.I. 0.34/0.56, *p* = 0.023, I^2^ = 19.57%, Het. *p*-value = 0.251). The operative time for laparoscopic surgery was shorter than that for robotic surgery (N = 2819, OR = 43.45, 95% C.I. 17.55/69.36, *p* < 0.001, I^2^ = 98.362%, Het. *p*-value < 0.001), even if the analysis showed a significative heterogeneity. Robotic-assisted surgery has a shorter time to the first flatus (Figure 4d (N = 1918, OR = 0.56, 95% C.I. −0.55/−0.09, *p* = 0.006, I^2^ = 84.264%, Het. *p*-value < 0.001) with a significative heterogeneity between the studies. No difference was found in length of hospitalization.

Figure 5 reports the analysis of estimated blood loss, quality of life, mortality rate, pain at one day and one month after surgery. The economic costs (Figure 5c) were higher in the robotic-assisted approach than in laparoscopic surgery (N = 193, OR = 1947.09, 95% C.I. 1136.62/2757.57, *p* < 0.001, I^2^ = 0.00%, Het. *p*-value = 0.705). The analysis of pain scores after one month shows no difference. No differences were found in estimated blood loss, mortality rate, quality of life at 1 month, pain at 1 day, and 1 month after the surgery. The outcomes quality of life and pain at 1 month were measured only in hernia repair surgery.

Appendix A shows the sensitivity analysis of conversion to open surgery rate, operative time, length of hospitalization, time to the first flatus, anastomotic leakage, readmission rate, and estimated blood loss. The sensitivity analysis confirmed the higher incidence of conversion to open surgery (Appendix A) (N = 2702, OR = 0.58, 95% C.I. 0.38/0.90, *p* = 0.013, I^2^ = 8.896%, Het. *p*-value = 0.360) and shorter operative time (Appendix A) (N = 2723, OR = 46.45, 95% C.I. 18.13/74.76, *p* < 0.001, I^2^ = 98.524%, Het. *p*-value < 0.001) in the laparoscopic group; the time to the first flatus (Appendix A) is shorter in robotic surgery (N = 1873, OR = −0.29, 95% C.I. −0.52/−0.05, *p* = 0.019, I^2^ = 85.974%, Het. *p*-value < 0.001). No differences were found in the sensitivity analysis for the other outcomes. Appendix A shows the subgroup analysis of secondary outcomes. Operative time results are longer only in robotic-assisted hernia repair (Appendix A) (N = 198, OR = 59.29, 95% C.I. 53.00/65.57, *p* < 0.001, I^2^ = 0%, Het. *p*-value = 0.548). The time to first flatus was shorter in robot-assisted gastrectomy (N = 519, OR = 0.50, 95% C.I. −0.65/−0.35, *p* < 0.001, I^2^ = 0.000%, Het. *p*-value = 1). No differences were found in the other outcomes in the different types of surgery.

### 3.3. Trial Sequential Analysis

Appendix A shows the TSA of the overall, sensitivity and subgroup comparisons for intra- and post-operative complications. The TSA evaluates the conclusiveness of the results of each comparison between the two approaches by plotting a cumulative z-score on a graph: if the z-score reaches the required information size and ends out the monitoring zone, the analysis for the considered outcome is conclusive and no more studies are needed. The metanalysis was conclusive only for the following analyses: the overall analysis of complications (Appendix A, required information size = 3201), the overall analysis for anastomotic leakage (Appendix A, required information size = 758), and sensitivity analysis of the anastomotic leakage (Appendix A, required information size = 758). The other analyses for complications rate were inconclusive. Appendix A reported the TSA for the rate of conversion to open surgery, operative time, length of hospitalization, and time to first flatus. Our overall and sensitivity analyses were not conclusive for conversion to open surgery rate, operative time, length of hospitalization, and time to first flatus. Appendix A shows the TSA of estimated blood loss, quality of life, mortality rate, pain after 1 day, and pain after 1 month. Only the overall and sensitivity analyses of estimated blood loss (Appendix A, required information size = 199) and the analysis of estimated blood loss in rectal surgery (Appendix A, required information size = 199) reached the required information size. The analysis is unconclusive for quality of life, mortality rate, pain after 1 day, and pain after 1 month because these analyses did not reach the required information size.

### 3.4. Fragility Index

Appendix A shows the fragility indexes (FIs) for each dichotomous outcome: their median, minimum, and maximum. No studies showed a robust median of FI in their own outcomes [22]. Only one study has an FI of 8 in deep vein thrombosis and mortality [34]; one study has a median FI of 8 for mortality and other respiratory complications [40]; one study has an FI of 8 in readmission rate [37].

## 4. Discussion

Our metanalysis highlighted that robotic surgery presents a lower incidence of pneumonia, need of conversion to open surgery, and a shorter time to first flatus after surgery than video-laparoscopic surgery; however, the robotic surgery seemed to be related with higher costs and longer operative time. Our work suggested that the robot-assisted and laparoscopic-assisted surgery do not differ in anastomotic leakage, cardiovascular and respiratory complications, readmission, and deep vein thrombosis; moreover, no differences were found in estimated blood loss, mortality, quality of life, and post-operative pain.

Pneumonia is an important post-operative complication with a dramatic impact on length of stay, morbidity, mortality, and economic costs in gastrointestinal surgery [42]. When pneumonia becomes complicated, especially in weak patients such as the elderly, its treatment involves the recovery in the intensive care unit, with a higher risk of other complications and mortality [43]. The pneumonia incidence seems to be related also to the gastrectomy procedure, diabetes, nursing care and other factors [44,45,46]. The different incidence of pneumonia between robotic and laparoscopic approaches could be explained by the lower pneumoperitoneum pressure of robotic surgery with a consequent lower extent of lung atelectasis [47]; indeed, the increasing of intrabdominal pressure due to carbon dioxide insufflation not only causes the lifting of diaphragm but also the increase in lung resistance and a rise in thoracic pressure with an augmented risk of pulmonary aspiration and hypoxemia [47]. Notably, other crucial risk factors for post-operative pulmonary complications are strictly related to the anesthesia management; for instance, a high concentration of inspired oxygen, an inappropriate neuromuscular blockade, and different ventilation strategies could lead to an augmented incidence of post-operative respiratory issues: more studies are needed to clarify the different contributions of anesthesia and surgery related factors to pulmonary complications [7,47,48,49,50].

The use of robotic surgery was found to have a lower conversion rate to open surgery compared to laparotomy; this result is in line with other meta-analyses which included randomized and non-randomized prospective studies [51,52]. The lower incidence of conversion to laparotomy could be explained by many factors such as a better view of the surgical field, the finest movements of robotic arms and increased degree of movement, with a consequent lower incidence of intraoperative injury of other structures [2].

The laparotomy approach can extend the length of hospitalization and mortality compared to the robotic and laparoscopic-assisted surgery [53,54]; a longer length of hospital stay is strictly related to an increasing of the risk of acquiring hospital infections which could lead to a higher mortality rate. A longer length of hospitalization can be due to many factors, and one of the most important is post-operative pain, which leads to a delayed recovery of bowel movements and a delayed mobilization with an increased risk of deep vein thrombosis, and pulmonary complications due to difficult breathing and lower thoracic expansion [55]. However, the lower incidence of conversion rate to open surgery could have a high impact on the economic costs by affecting the post-operative morbidity and mortality of the patients; therefore, a deep economic cost–benefits analysis is needed to clarify the real advantage of the robotic approach.

The analysis showed a longer operative time in robotic surgery especially in hernia repair. The difference could be explained by the time to place the trocar and the arms of the robot, even if our results suggested the equal duration of colon–rectal surgery [56].

The time to the first flatus, which indicates the recovery of bowel motility, is shorter in robot-assisted colon–rectal surgery. Even if the causes of post-operative ileus are not totally known, the post-operative ileus seems to be associated with intestinal manipulation during surgery [57]; indeed, the manipulation of the bowel during the surgery seems to cause an increasing of cholinergic tone and a liberation of inflammatory molecules which stop the bowel movements [57]. The finest movements of robotic arms and the magnification of the surgical field could cause a lower manipulation of the intestine, thus reducing the time needed for bowel motility recovery. Even if other studies suggested that the length of stay is shorter in patients undergoing robotic rather than laparoscopic surgery, we did not find a difference between the two approaches [2,53].

Our results showed that the estimated blood loss, mortality, quality of life, and pain control are similar between the two approaches. Even if there is a difference between open surgery and laparoscopic or robotic-assisted surgery, with a lower estimated blood loss in mini-invasive approaches, there is no clinically relevant advantage between the mini-invasive approaches which we investigated in our metanalysis [2]. The similar quality of life in hernia repair between the two approaches is not in line with other articles that suggested that the quality of life is higher in robotic surgery in comparison with laparoscopic and open techniques [58]. According to our other systematic review and metanalysis, pain control seems to be the same, but according to the TSA, our analysis was not conclusive, and more studies are needed to assess this outcome in hernia repair and other gastrointestinal surgical procedures [52].

It is important to underline that robotic-assisted surgery is related with a higher economic cost than laparoscopic surgery. The evaluated costs of robotic surgery are limited to hospital stay and the cost of the devices; it does not consider the social and psychological impact on the patient. The lack of the psychological, social, and long-term outcomes in the studies so far conducted about robotic surgery hinders a comprehensive evaluation of the economic impact of this technique and, in addition, an overarching analysis is needed to clarify the effect of the lower rate of conversion, operative time, and time to first flatus on the overall costs.

Our work has some limitations: the number of studies is not sufficient for an exhaustive analysis of every outcome we considered for the different types of gastrointestinal surgery; some outcomes, such as pain and quality of life, were evaluated only in hernia repair; the enrolled studies are characterized by a low intrinsic robustness; it was not possible to consider and evaluate the anesthesia management and its impact on outcomes such as pneumonia; the majority of the studies on robotic surgery were conducted with Da Vinci^®^ and no sufficient data are available about other robotic systems; the heterogeneity of some analyzed studies reduces the reliability of our results. It is important to underline that our analysis is conclusive only for few outcomes, such as anastomotic leakage and estimated blood loss, and this could explain the discrepancy between our findings and other similar works; as the TSA suggested, more studies and high-quality RCTs are needed to define the different impact of the laparoscopic and robotic surgery on the considered outcomes.

## 5. Conclusions

The robotic-assisted gastrointestinal surgery has a lower incidence of pneumonia, a lower conversion rate to open surgery, and an earlier post-operative recovery of bowel motility than the laparoscopic approach; therefore, the use of robotic surgery is limited by a higher economic impact and a longer operative time, with no significant advantages on blood loss or post-operative complications, such as deep vein thrombosis, anastomotic leakage, and cardiovascular complications, compared to the laparoscopic approach. More studies are needed to clarify the impact of robot-assisted surgery on intra- and post-operative complications for the different types of gastrointestinal surgeries; moreover, comprehensive cost-effectiveness studies with a randomized-controlled design should be conducted to establish if robotic-surgery is more effective in reducing costs compared to both the laparoscopic and open surgery techniques.

## Figures and Tables

**Figure 1 jpm-13-01297-f001:**
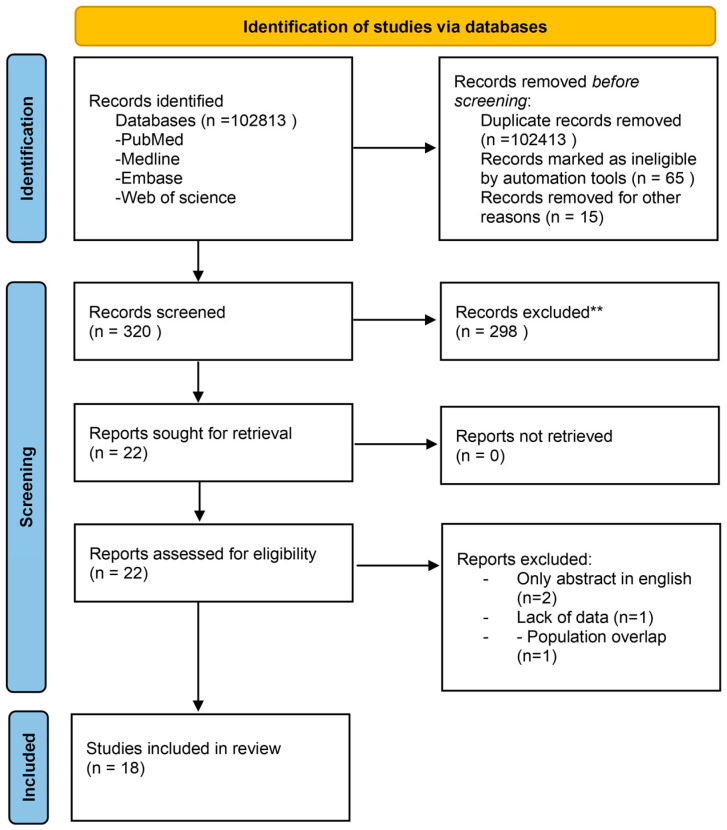
PRISMA flowchart. Identification, screening, and evaluation process of the included studies.

**Figure 2 jpm-13-01297-f002:**
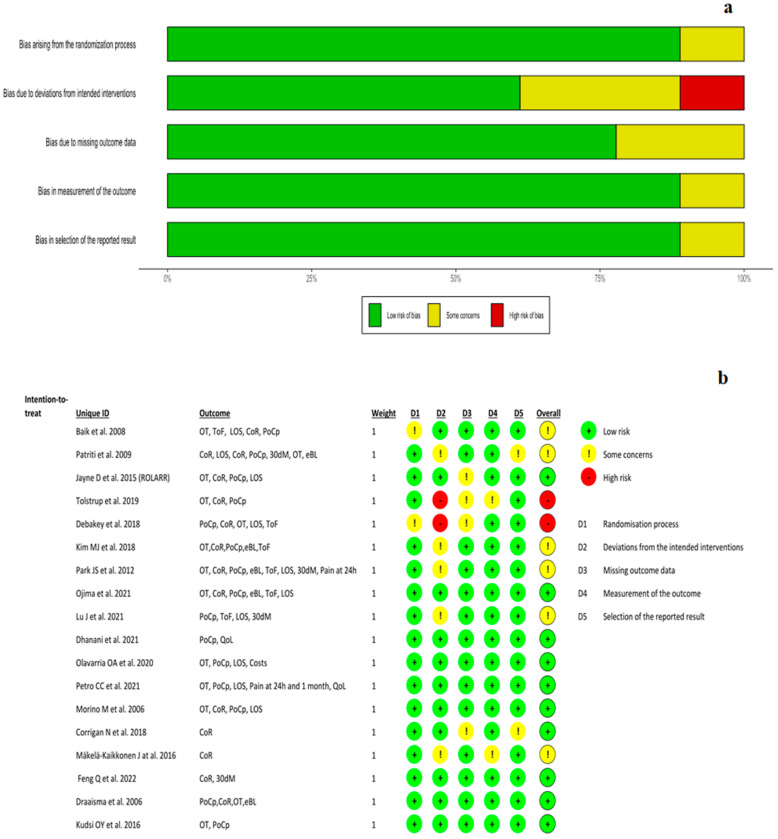
Risk of bias. (**a**) summary plot; (**b**) summary figure [9,25,26,27,28,29,30,31,32,33,34,35,36,37,38,39,40,41].

**Figure 3 jpm-13-01297-f003:**
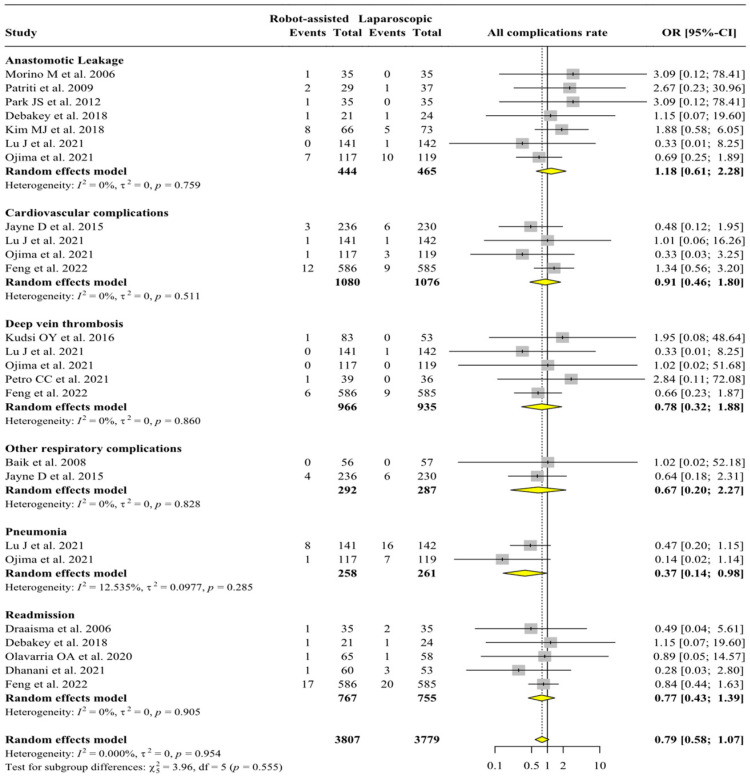
Analysis of all post-operative complications. Overall analysis of intra- and post-operative complications. From the top: anastomotic leakage rate [9,25,30,33,34,39,40]; cardiovascular complications rate [9,31,37,40]; deep vein thrombosis rate [9,28,37,40,41]; other respiratory complications rate [29,31]; pneumonia rate [9,40]; readmission rate [26,27,33,37,38]; overall analysis [9,25,26,27,28,29,30,31,33,34,37,38,39,40,41].

**Figure 4 jpm-13-01297-f004:**
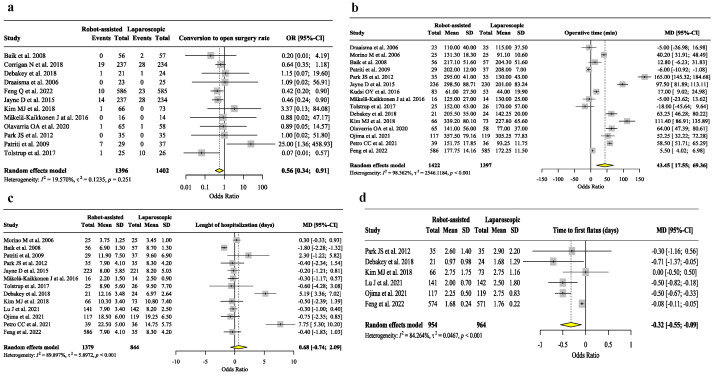
Analysis of conversion to open surgery rate, operative time, length of hospitalization, and time to first flatus. (**a**) Analysis of conversion to open surgery rate [25,27,29,30,31,32,33,34,35,36,37,38]; (**b**) analysis of operative time (in minutes) [25,27,28,29,30,31,32,33,34,36,37,38,39,40,41]; (**c**) analysis of length of hospitalization (in days) [9,25,28,29,30,31,32,33,34,36,37,39,40]; (**d**) analysis of time to first flatus (in days) [9,25,33,34,37,40].

**Figure 5 jpm-13-01297-f005:**
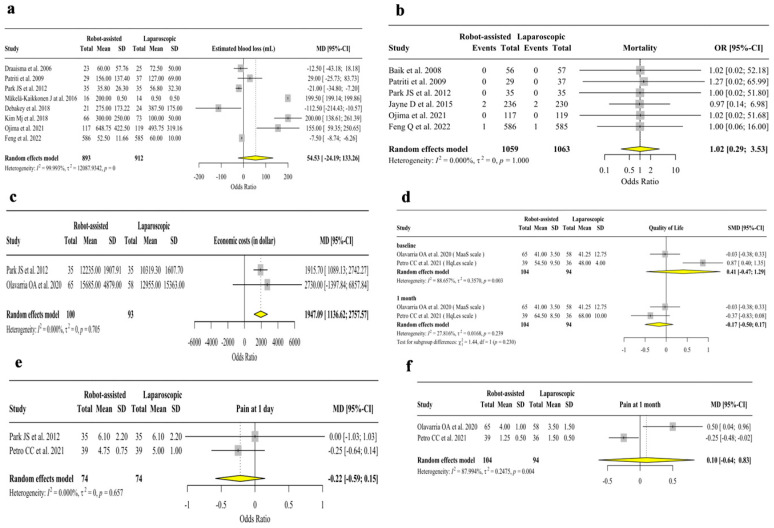
Analysis of estimated blood loss, mortality rate, economic costs, quality of life score at 1 month, pain score at 1 day and 1 month. (**a**) Analysis of estimated blood loss (in mL) [25,30,33,34,36,37,38,40]; (**b**) analysis of mortality rate) [25,29,30,31,37,40]; (**c**) analysis of economic costs (in dollar) [25,27]; (**d**) analysis of quality of life scores after 1 month [27,28]; (**e**) analysis of pain score at one day after surgery [25,28]; (**f**) analysis of pain scores one month after surgery [27,28].

## Data Availability

Data presented are contained within the article; for additional information, datasets are also available upon request from the corresponding author.

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
