# Peer review of "Robot-Assisted versus Laparoscopic Gastrointestinal Surgery: A Systematic Review and Metanalysis of Intra- and Post-Operative Complications"

_jpm, 2023, doi:10.3390/jpm13091297_

Round 1

Reviewer 1 Report

This is a well written metaanalysis regarding the intra-postoperative complications of minimally invasive surgeyr in gastrointestinal area.

the article and methodology is god. the scientific language is ok.

Author Response

Point 1: This is a well written metaanalysis regarding the intra-postoperative complications of minimally invasive surgeyr in gastrointestinal area.

the article and methodology is god. the scientific language is ok.

Response 1

 We really appreciate your comment on our work, and we are so grateful for the time that you spent reading this manuscript.

Reviewer 2 Report

The study investigated the impact of robot systems on postoperative complications, length of hospitalization, estimated blood loss, quality of life, conversion to open surgery rate, and economic costs in comparison to the video-laparoscopic assisted technique in elective gastrointestinal surgery, and showed that the robotic gastrointestinal surgery has some advantages compared to laparoscopic technique such as lower conversion rate, faster recovery of bowel movement, but it results higher economic costs. Following are some concerns that the authors need to revise and address before considering publication.

For the introduction part:

1. Please add the references in the first paragraph.

2. Line 45, please check the percentage of laparoscopic abdominal GI procedures.

3. Please add information on what are the contributions of the study.

For the material and method part:

4. Please check the format of figure 1 and revise it.

For the result part:

5. The authors need to revise figures 2, 3, 4, 5. These figures are very low quality, and the reader cannot see the content clearly.

6. In lines 211 - 214; 222 - 224, please double-check grammar (tense) and revise them.

7. What are the purpose of TSA in the study?

For the discussion part:

8. Please add the discussion on the result of trial sequential analysis.

can be edited better.

Author Response

Comments and Suggestions for Authors

The study investigated the impact of robot systems on postoperative complications, length of hospitalization, estimated blood loss, quality of life, conversion to open surgery rate, and economic costs in comparison to the video-laparoscopic assisted technique in elective gastrointestinal surgery, and showed that the robotic gastrointestinal surgery has some advantages compared to laparoscopic technique such as lower conversion rate, faster recovery of bowel movement, but it results higher economic costs. Following are some concerns that the authors need to revise and address before considering publication.

For the introduction part:

Point 1. Please add the references in the first paragraph.

Response 1

We really appreciate the suggestion of the reviewer. We modified the first paragraph of the introduction adding some new citations and updating the references.

Page 1; Lines 36,39

<< Robotic surgery is gaining an ever-growing interest as it allows surgeons to have a three-dimensional vision of the operating field, fine movements of robotic arms, and smaller incisions, leading to many advantages such as reduced blood loss, lower risk of damage of nervous, vascular, and parenchymal structures, and less postoperative pain, but it is also associated with extreme position of the patient that can affect the incidence of some side effects, such as atelectasis[1–3]. Even if robotic surgery shares many characteristics with standard laparoscopic technique, it overcomes some drawbacks of laparoscopy such as restricted range of motion of instruments, poorly ergonomic position of the surgeon, and the two-dimensional view[1,4].>>

Page 14; Lines 567-575

<<1.      Baukloh, J.-K.; Perez, D.; Reeh, M.; Biebl, M.; Izbicki, J.R.; Pratschke, J.; Aigner, F. Lower Gastrointestinal Surgery: Robotic Surgery versus Laparoscopic Procedures. Visc Med 2018, 34, 16–22, doi:10.1159/000486008.

  1. Palep, J. Robotic Assisted Minimally Invasive Surgery. J Minim Access Surg 2009, 5, 1, doi:10.4103/0972-9941.51313.
  2. Luca, F.; Valvo, M.; Ghezzi, T.L.; Zuccaro, M.; Cenciarelli, S.; Trovato, C.; Sonzogni, A.; Biffi, R. Impact of Robotic Surgery on Sexual and Urinary Functions After Fully Robotic Nerve-Sparing Total Mesorectal Excision for Rectal Cancer. Ann Surg 2013, 257, 672–678, doi:10.1097/SLA.0b013e318269d03b.
  3. Bencini, L.; Annecchiarico, M.; Di Marino, M.; Moraldi, L.; Perna, F.; Coratti, A. Gastrointestinal Robotic Surgery: Challenges and Developments. Robotic Surgery: Research and Reviews 2015, 11, doi:10.2147/RSRR.S50266.>>

Point 2. Line 45, please check the percentage of laparoscopic abdominal GI procedures.

Response 2

Thanks to the reviewer for his/her comment. We edited the line as suggested.

Pages 1-2; Lines 44-46

<< [..] a recent study reported that the average number of the laparoscopic abdominal GI procedures performed by residents increased considerably in the last decades[6]>>

Point 3. Please add information on what are the contributions of the study.

Response 3

We added a sentence at the end of introduction that summarizes the contributions of our work.

Page 2, Lines 77-85

<<To our knowledge, this is the first systematic review and metanalysis that investigated the incidence of intra- and post- operative complications between the laparoscopic and robotic assisted approach. The different incidence of pneumonia of robotic-assisted surgery should be considered in the decision-making process to evaluate which approach could be used, especially in patients with high risk factors for pneumonia undergoing elective GI surgery. Moreover, it is unclear the clinical impact of the lower rate of conversion to open surgery and the faster recovery of the bowel motility of the robotic surgery and more studies are needed to define the real advantages and costs of the robotic-assisted GI surgery.>>

For the material and method part:

Point 4. Please check the format of figure 1 and revise it.

Response 4

 Thanks to the reviewer for his/her comment. We edited as suggested the figure 1.

Page 5; Figure 1.

For the result part:

Point 5. The authors need to revise figures 2, 3, 4, 5. These figures are very low quality, and the reader cannot see the content clearly.

Response 5

We improved the quality of the figures as suggested.

Page 6; Figure 2.

Page 7; Figure 3.

Page 8; Figure 4, Figure 5.

Point 6. In lines 211 - 214; 222 - 224, please double-check grammar (tense) and revise them.

Response 6

Thanks to the reviewer for pointing this out. We revised the sentence as follow:

Page 7, Lines 225-228

<< The operative time for laparoscopic surgery was shorter than for robotic surgery(N= 2819, OR= 43.45, 95% C.I. 17.55/69.36, p<0.001, I2=98.362%, Het. P-value<0.001) , even if the analysis showed a significative heterogeneity.>>

Page 8, Lines 238-240

<< The economic costs were higher in robotic assisted approach than laparoscopic surgery>>

Point 7. What are the purpose of TSA in the study?

Response 7

Thanks to the reviewer for giving us the possibility to explain better our work. We clarify what results are inconclusive to establish a clear impact on the analyzed outcomes between the considered approaches.

Page 9. Lines 267-271

<<The TSA evaluates the conclusiveness of the results of each comparison between the two approaches by plotting a cumulative z-score on a graph: if the z-score reaches the required information size and ends out the monitoring zone, the analysis for the considered outcome is conclusive and no more studies are needed.>>

For the discussion part:

Point 8. Please add the discussion on the result of trial sequential analysis.

Response 8

Thanks to the reviewer for pointing this out. We added at the end of the discussion   a summary of the final interpretation of trial sequential analysis.

Page 11, Lines 378-383

<< It is important to underline that our analysis is conclusive only for few outcomes, such as anastomotic leakage and estimated blood loss, and this could explain the discrepancy between our findings and other similar works; as TSA suggested, more studies and high-quality RCTs are needed to define the different impact of the laparoscopic and robotic surgery on the considered outcomes.>>